# Comparison of Wing, Ovipositor, and Cornus Morphologies between *Sirex noctilio* and *Sirex nitobei* Using Geometric Morphometrics

**DOI:** 10.3390/insects11020084

**Published:** 2020-01-24

**Authors:** Ming Wang, Lixiang Wang, Ningning Fu, Chenglong Gao, Tegen Ao, Lili Ren, Youqing Luo

**Affiliations:** 1Beijing Key Laboratory for Forest Pest Control, Beijing Forestry University, Beijing 100083, China; 13020028768@163.com (M.W.); funingning2012@sina.com (N.F.); gaocl890907@163.com (C.G.); 2Sino-French joint Laboratory for Invasive Forest Pests in Eurasia, INRA-Beijing Forestry University, Beijing 100083, China; 3College of Plant Protection, Gansu Agricultural University, Lanzhou 730070, China; wanglx@gsau.edu.cn; 4Tongliao Control and Quarantine Station of Forest Pest, Tongliao 028000, China; autogen1@163.com

**Keywords:** cornus, geometric morphometrics, ovipositor, Siricidae, taxonomy, wing

## Abstract

*Sirex noctilio* F. (Hymenoptera: Siricidae) is an invasive woodwasp from Europe and North Africa. Globalization has led to an expanding global presence in pine forests. *S. noctilio* has been previously introduced outside of its native range and now co-occurs in trees with native *S. nitobei* Matsumura (first discovered in 2016). Damage to *Pinus sylvestris* var. *mongolica* Litv in northeast China can be attributed to two types of woodwasp. To distinguish the two species by the traditional taxonomic morphology, we mainly differentiate the color of the male’s abdomen and the female’s leg. There remains intraspecific variation like leg color in the delimitation of related genera or sibling species of *Sirex* woodwasps. In this study, we used landmark-based geometric morphometrics including principal component analysis, canonical variate analysis, thin-plate splines, and cluster analysis to analyze and compare the wings, ovipositors, and cornus of two woodwasps to ascertain whether this approach is reliable for taxonomic studies of this group. The results showed significant differences in forewing venation and the shapes of pits in the middle of ovipositors among the two species, whereas little difference in hindwings and cornus was observed. This study assists in clarifying the taxonomic uncertainties of Siricidae and lays a foundation for further studies of the interspecific relationships of the genus *Sirex*.

## 1. Introduction

Siricidae is a small group of species, in which individuals are relatively large with a clean body surface and easily identifiable morphological features. One of the most striking features of Siricidae is what appears to be incredible variation in wing venation, including the appearance or the disappearance of veins symmetrically or asymmetrically on either wing. Such variation is very rarely seen in other Hymenoptera, a group where wing veins are important for classification [1]. The wing characteristics of Siricidae are unstable and seldom used in taxonomic studies. In general, the classification and identification of this group is based mainly on the structure of the thorax and abdomen. However, these structures are very similar in closely related species; thus, it is difficult to accurately identify species in some cases. Interestingly, geometrics, as a new classification method, has been recently applied in many classification studies of Hymenoptera [2].

It is well known that insect wing shape can exhibit a high heritability in nature [3,4]. Thus, wing morphology is of primary importance to entomologists studying systematics. Since the 1970s, several investigators have used the two-dimensional characteristics of insect wings to advance the fields of systematics and phylogeny [5,6]. Geometric morphometrics utilizes powerful and comprehensive statistical procedures to analyze the variations in shape using either homologous landmarks or outlines of the structure [6,7,8], and it is currently considered to be the most rigorous morphometric method. Wings are excellent in studies that define morphological variations because they are nearly two-dimensional and their venation provides many well-defined morphological landmarks [9]. For instance, vein intersections are easily identifiable, which enables the general shape of the wing to be captured [10]. The use of geometric morphometrics to study wing venation has also been useful in insect identification at the individual level [11], in differentiation between sibling species [12,13], and in delimitation among genera. However, thismethodology has not yet been applied in studies of woodwasps.

The wood-boring wasp, *Sirex noctilio* Fabricius (Hymenoptera: Siricidae), is an invasive pest of numerous species of pine tree (*Pinus* spp.) worldwide and most of the destruction from *S. noctilio* is in commercial plants [14,15]. In August 2013, woodwasps were detected as a pest of Mongolian pine (*Pinus sylvestris* var. *mongolica*) in the Duerbote Mongolian Autonomous County, Heilongjiang Province, China [16]. In 2016, two morphologically similar woodwasps were found to damage *P. sylvestris* var. *mongolica* in Jinbaotun forest farm, Inner Mongolia, causing a lot of pine forests to weaken and die. After morphological comparison and molecular identification, the two woodwasps were *S. noctilio* and *S. nitobei* [17]. Each have two pairs of large, transparent-film wings with visible mesh veins. As the insect wing is a planar structure, it is relatively easy to acquire two-dimensional images, and it is difficult to unintentionally distort the structure. Unfortunately, it is rather difficult to identify *S. noctilio* and *S. nitobei* wing vein characteristics with the naked eye, and these two species display unstable vein patterns, which means that the use of geometric morphometrics is appropriate [18].

In previous studies, the pits on the ventral portion (lancet) of the ovipositor have been consistently used as an identifying structure. The lancets of the ovipositor independently slide back and forth to move the egg and to penetrate wood. This characteristic was used for the first time by Kjellander (1945) to segregate females of *S. juvencus* from those of *S. noctilio*. Furthermore, the size, shape, and number of pits on the ovipositors can be used as distinguishing features for the identification of most species [15]. This also holds true for *Sirex*, in which the most important distinguishing characteristics on the ovipositor are pits located from the base to approximately the middle of the lancet, although the apical teeth segments usually do not show distinct differences [15]. Another striking diagnostic feature is the large hornlike projection, called the cornus, on the last abdominal segment of the females. The cornus is thought to help the larvae pack the frass in the tunnel. The cornus varies in shape (the shape of the female cornus does not vary with size for most species), although their distinguishing features remain poorly characterized. These difficulties underline the need for further studies to clarify the taxonomy of woodwasps, either by searching for new morphological characteristics with clear distinguishing variations or applying alternative effective methods to provide a basis for studying flight and reproductive behavior.

Geometric morphometrics [6,19,20] overcomes the shortcomings of conventional morphological analysis and focuses on the topological information of the organic form [18]. In addition, as it is not affected by various factors, such as size and shape, this method has the potential to be more widely used in the identification of insects, resulting in automatic insect recognition system that is continually updated and improved [3]. In taxonomy and other fields, genetics and morphometrics are complementary tools that are often used to understand the origins of phenotypic differences. The application of marker points in biology can be divided into three categories [21]. In this study, we focused on two of these categories, namely (i) the common points that can be accurately found on each specimen based on the anatomical features, which is a mathematical point supported by substantial evidence between homologous subjects [22], such as structural intersections (the basis for marking points on wings); and (ii) the mathematical point for homologous subjects, which is supported by geometrical rather than histological evidence, such as depressed or convex points (the basis for marking points on ovipositors and cornus). Platts analysis was used to superimpose the marker points and minimize the deviation of the marker points. In the same coordinate system, the influence of non-morphological factors in morphological information analysis was eliminated, and the average contour of each population was obtained. Thus, in the present study, we applied landmark-based geometric morphometrics to quantify and analyze wing, cornus, and ovipositor morphologies of two *Sirex* species that have not been previously characterized. We explored the similarities between these species to strengthen the available quantitative research data that form the basis of species identification and to provide new insights for automatic insect identification systems.

## 2. Materials and Methods

### 2.1. Ethics Statement

This study did not involve endangered or protected species. No specific permits were required for this study.

### 2.2. Insects

Insect samples were collected from the Jinbaotun Forest in Tongliao City, Inner Mongolia, from June 2016 to September 2017. Trees were felled in early summer, and insects were collected at emergence. Insects were collected from different, unrelated plots. Sirex specimens were structurally analyzed (Table 1). Prior to geometric morphometric analysis, specimens were identified using adult morphological characteristics, including the color of their thoracic legs, abdomen [1,23].

### 2.3. Insect Processing and Image Acquisition

The front and rear wings of each specimen were cut off from the body. Rohlf et al. suggested using only one side of each paired organ or limb to avoid asymmetry bias between the two halves [24]. In this study, only the left wings of specimens were used, which were ultrasonically cleaned with 75% alcohol for 90 s to remove impurities. Thereafter, specimens were dehydrated with an ascending series of ethanol washes (75%, 80%, 85%, 90%, 95%, 100%, and 100%) for 20 min each. Specimens were softened with xylene, placed on glass slides that were previously wiped clean with a lens tissue, and then mounted in neutral balsam.

Ovipositors and cornus were dissected from the *Sirex* specimens, and the remaining parts were stored at −20 °C. We used ten individuals of each species to examine the pits and cornus. Impurities were removed from the specimens by ultrasonic cleaning (Skymen, JP-1200, Shenzhen, China) or brushing. The specimens were then placed face up on clean glass slides and mounted in neutral gum (Coolaber, Beijing, China). All specimens were numbered.

A light microscopy (Leica, S4E, Wetzlar, Germany) was performed to determine the number and distribution of pits on ovipositors. Images were captured with a Nikon camera (Nikon D90, Tokyo, Japan). Each image was saved as a 24-bit. bmp image, and original stored images were used in subsequent analysis rather than compressed files. The directions and positions of the specimen images were readjusted with Photoshop CC2015 software (Adobe Systems, San Jose, USA).

### 2.4. Standardization of Data and Statistical Analysis

The TPS files were maked from selected images of wings, ovipositors, and cornus using TpsUtil software (tpsUtil 1.47, [24]). The landmarks in each image were recorded as the central location point of each specimen and digitized using TpsDig2 software (New York, NY, USA) [24]. For each. Tps file, the landmarks were scanned in the same order, and the scale factor was set for each image. Therefore, 20 landmarks from the forewing, 11 landmarks from the hindwing (Figure 1a), nine landmarks (Figure 2) from the ovipositor (from the base to the middle pits, Nos. 14, 15, 16), and five landmarks from the cornus (Figure 3) were digitized.

After the. Tps files were converted into nts files using TpsUtil software and the marker information was saved, the images were processed with MorphoJ software [25,26]. The variations in shape were assessed by principal component analysis. To better visualize the variations in shape, we determined the average configuration of landmarks for each species. Deformation grids were used to show the variations. The relative similarity and discrimination of the species was analyzed using canonical variate analysis, which identified changes in shape using mean values of the two groups by assuming that covariate matrices were identical [27]. Canonical variate analysis is a reliable method for identifying differences among taxa. Procrustes ANOVA (Analysis of Variance) [25,28] was utilized to determine significant differences among species [29]. Furthermore, PAST software [30,31] was used to generate phenograms by cluster analysis that utilized Euclidean distances calculated from the matrix of the Procrustes shape coordinates. ImageJ software [32] was used to calculate the wing area. All images were converted into binary files, and the background was removed, which resulted in a black wing surrounded by a white space (Figure 1c). The wing outline was assessed, and minor damage to the wing outline was eliminated. The pixels per mm were calculated using a ruler of known scale, and the wing area was obtained.

## 3. Results

### 3.1. Shape Variables of the Wings in the Genera of S. noctilio and S. nitobei

#### 3.1.1. Analysis of Female Forewings

Principal component analysis showed shape variations in *S. noctilio* and *S. nitobei* wings (Figure 4a). The results of Procrustes ANOVA explained 42.79% of the intergroup variations in *S. noctilio* and *S. nitobei* female forewings. Significant differences in the forewings were observed between the two species by principal component analysis (Figure 4a) and cluster analysis (Figure 4d). Mahalanobis distances between *S. noctilio* and *S. nitobei* female wings are significantly different in comparisons (*p* < 0.05), and Procrustes distances (*p* < 0.05) are similar (Table 2).

The lollipops and deformation grids indicated the direction and magnitude of the shape variations by principal component analysis (Figure 4b) and canonical variate analysis (Figure 4c). The deformation grids of the first between-group principal component revealed differences in the junction (No. 5) of Cu and 2m-cu, the junction (No. 16) of R1 and Rs2, the vannal region (Nos. 1, 3, 4) of 2A and 2cu-a, 2A and a, 1A and a. The deformation grids of the second between-group principal component revealed differences in the Rs and 2r (No. 19) and the region (around junctions Nos. 10, 17, 2). The deformation grids of the first between-group canonical variate showed that contributing most to the shape differences between them was the junctions of Cu and 2m-cu etc. (Nos. 5, 16, 3, 4, 12) (Figure 4c) (Table 3).

#### 3.1.2. Analysis of Female Hindwings

The results of Procrustes ANOVA explained 43.71% of the intergroup variations in *S. noctilio* and *S. nitobei* female hindwings. There was also significant difference between two groups in principal component analysis (Figure 5a) and cluster analysis (Figure 5d). The deformation grids of the first principal component revealed differences in the junction (No. 2) of 1A and 2A (Figure 5b). Also, the junctions of Cu and m-cu, Cu and cu-a, C (costa) and R1 (Nos. 3, 4, 11) appeared variable in species, whereas those of the second principal component revealed differences in the remigium (Nos. 7, 5, 11, 2, 8, 10) (Figure 5b). The deformation grids of the first canonical variate showed significant differences in the region of vannal fold (Nos. 2, 3, 4) (Figure 5c) (Table 4).

#### 3.1.3. Analysis of Male Forewings

The results of Procrustes ANOVA explained 35.29% of the intergroup variations in *S. noctilio* and *S. nitobei* male forewings. Significant differences in Mahalanobis distances of the male wings were observed between the two species (*p* < 0.0001, Table 5). These findings were consistent with those of principal component analysis (Figure 6a) and cluster analysis (Figure 6d or Figure 7d).

The deformation grids of the first principal component revealed differences in the remigium (No. 16), the junction of Cu and 2m-cu, 2A and a etc. (Nos. 5, 3, 4, 18,13) (Figure 6b). These findings were similar to the deformation grids of the first canonical variate (Figure 6c). The deformation grids of the second principal component revealed differences in the junction of M and 2m-cu, 2A and 2cu-a, M and Cu1 (Nos. 12, 1, 10) (Figure 6b).

#### 3.1.4. Analysis of Male Hindwings

The results of Procrustes ANOVA explained 32.89% of the intergroup variations in *S. noctilio* and *S. nitobei* male hindwings. The principal component analysis of the two species had individual sample overlap.

The deformation grids of the first principal component revealed differences in the region (around by junctions Nos. 2, 4, 9, 8) (Figure 7b) were similar to the deformation grids of the first canonical variate (Figure 7c), whereas those of the second principal component revealed differences in the junctions of 1A and 2A, M and 1r-m, Cu and cu-a (Figure 7b).

#### 3.1.5. The Relationship between Wings and Dry Weight of Sirex

No significant differences were observed in the hindwings of the two woodborers. The total forewing area of male and female *S. noctilio* adults was significantly different from that of *S. nitobei* adults (F = 19.12; df = 3, 36; *p* < 0.0001; Figure 8). There was a positive correlation between the dry weight and forewing length between the two species (*S. noctilio*: r = 0.8588; *p* < 0.0001; S. *nitobei*: r = 0.8837; *p* <0.0001; Figure 9).

### 3.2. Shape Variables of the Ovipositors in the Genera of S. noctilio and S. nitobei

The results of Procrustes ANOVA explained 67.94% of the intergroup variations in *S. noctilio* and *S. nitobei* female ovipositors. Mahalanobis distances between the two species were significantly different in pairwise comparisons (*p* < 0.0001), and Procrustes distances were similar (*p* < 0.0001) (Table 6). These findings were confirmed by the results of principal component analysis (Figure 10a) and cluster analysis.

The deformation grids of the first principal component (Figure 10b) and first canonical variate (Figure 10c) revealed differences in the angle (1, 4, 7) of the average pit, whereas those of the second between-group principal component revealed differences in the bottom left points (2, 8).

### 3.3. Shape Variables of the Cornus in the Genera of S. noctilio and S. nitobei

The results of Procrustes ANOVA explained 31.24% of the intergroup variations in *S. noctilio* and *S. nitobei* cornus. Mahalanobis distances among the two species were significantly different in pairwise comparisons (*p* < 0.05), and Procrustes distances were similar (*p* < 0.05) (Table 7). These findings had differences in those of cluster analysis (Figure 11d). Cluster analysis (Figure 11d) revealed that several individuals (e.g., 9, 10, 11) could not be clustered, which might have been due to differences in the tunnel environment and ossification structure. In general, the results of quantitative geometric analysis were consistent with those obtained by the naked eye.

## 4. Discussion

In recent years, we have witnessed monumental improvements in geometric morphometrics, which have enhanced the study of insect morphology. In general, these methods separate species shape from size and primarily focus on the shape as the key morphological characteristic, as few variables can reveal morphological differences between similar species. Geometric measurement methods, together with relevant mathematical models (e.g., principal component analysis, canonical variate analysis, and cluster analysis), can be used to acquire information on the morphology, genetic differentiation, development, and behavior of insects. When competing for resources on the same part of the host plant, the morphological structure of the two woodwasps may occur adaptive genetic variation. The use of geometric morphometrics allowed us to explain morphological similarities and differences between the two *Sirex* species, of which thin-spline analysis plots showed average contour distortion, and the length of the stick demonstrated the change size.

Compared with conventional classification methods, geometric survey methods can identify subtle morphological differences in insects [33]. To distinguish the two species from the traditional taxonomic morphology, we mainly distinguish the color of the male’s abdomen and the female’s leg. However, there is intraspecific variation in color patterns on the legs, abdomen and antennae, for example, females of *Sirex californicus*, *S. nitidus*, and *S. noctilio* each have pale and dark leg color morphs [34,35]. The results of relative warp analysis can show differences in the classification of intraspecies and interspecies [36]. In studies of morphology, the small unit on the wing is usually independent unit in the shape change, and has a certain genetic basis. Thus, different insects have different wing types and wing vein structures [3]. However, in similar species, these differences may be indistinguishable, with minor variations in the direction and branching of veins [37]. The shapes and vein profiles of insect wings contain valuable information, although it is difficult to understand the effects of behavioral and environmental factors on morphological variations using conventional classification methods [33].

Insect flight involves the first two branches of the radius vein and the thicker area of the wing film [38]), which is gradually reduced from the base to the end of the wing. *S. noctilio* woodwasps have a variable flight behavior, which relates to initial body size [39]. The body size of the *Sirex* also varies in different regions, and the two woodwasps cannot be distinguished simply by it. *Sirex* species mainly use carbohydrates for fuel during flight. The dry weight of *S. noctilio* is significantly larger than *S. nitobei* in this study. Therefore, the wing variations of both *Sirex* species in this study might explain the differences in flight ability and behavior. In addition, the ovipositor, an appendage through which females deposit eggs, plays important roles in sensing the microenvironment and initiating the laying of eggs. In this study, we observed differences in forewing and ovipositor shape, which were due to developmental plasticity. These findings provide a strong basis for further research on flight behavior and ovipositor function in various species.

In this study, we identified and classified two *Sirex* species, although the lack of inclusion of other Siricidae species was a major limitation. When species cannot be immediately identified by their appearance, landmark points can be easily extracted and analyzed. We anticipate that additional landmark points, such as those of the head, chest, and other parts, will be used in future studies [40]. Geographic distance is one of the key factors of population differentiation for widespread species. It is generally believed that the more geographically separated populations have less chance of gene exchange, resulting in morphological differences between populations. *S. noctilio* has invaded many areas in China, next we can collect samples from different regions for analysis of different geographical populations. At the same time, this paper provides basic data for the automatic recognition system of insects. Further identification of *Sirex* species will be provided. In addition, insects have a short life cycle and a fast response to the environment. Using geometric morphology can accurately analyze small changes in morphological structures and possible evolutionary trends in a short period of time.

## 5. Conclusions

There is species variation in the wing veins of two woodwasps. We selected the homology coordinate points for analysis. In conclusion, there were significant differences in the forewings and the pits on the ovipositor between the invasive *S. noctilio* and the native *S. nitobei*. The results showed that the taxonomic importance of hind wing venation and cornus characters was not stable for the two woodwasps. Geometric morphology can be used for morphological identification of insects of the genus *Sirex*, especially those species with variable coloration. We can distinguish the two woodwasps from the flight-related forewing veins and the reproduction-related ovipositor pits. Comparing the invasive species with their congeners can partialy avoid the bias due to taxonomical relatedness and enhance the credibility of the results. With landmark-based geometric morphometrics to quantify and analyze wing, cornus, and ovipositor morphologies of two *Sirex* species, we provide new insights for automatic insect identification systems. The currently used approach to study the morphology of wings is complicated and time consuming. This process may damage wings and several software packages cannot extract landmark points. In these cases, the user must use a computer mouse to manually select landmarks, as was performed in this study; however, measurements are affected by human factors. Further studies are needed to expand the identification system of insects by including different types of insects and performing different types of geometric morphometric analysis. New insect identification software packages should also be developed, as they can reduce the repetitive workload for investigators working in agriculture, forestry, quarantine, and other front-line industries.

## Figures and Tables

**Figure 1 insects-11-00084-f001:**
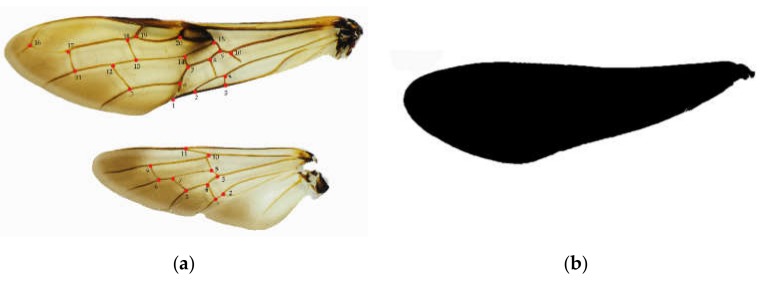
Description of the landmarks used in geometric morphometric analysis. (**a**) Locations of the 20 landmarks on the forewing of *Sirex* considered in the geometric morphometric analysis, locations of the 11 landmarks on the hindwing of *Sirex* considered in the geometric morphometric analysis; (**b**) The black wing image which was converted to binary.

**Figure 2 insects-11-00084-f002:**
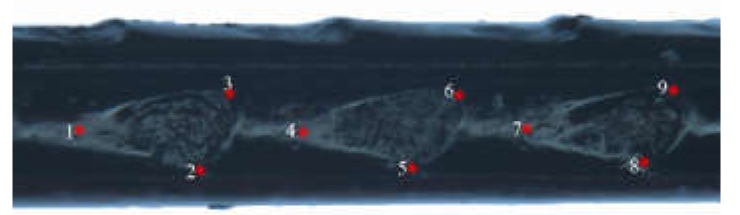
Position of 9 type Џ landmarks on the ovipositor of *Sirex* considered in the geometric morphometric analysis.

**Figure 3 insects-11-00084-f003:**
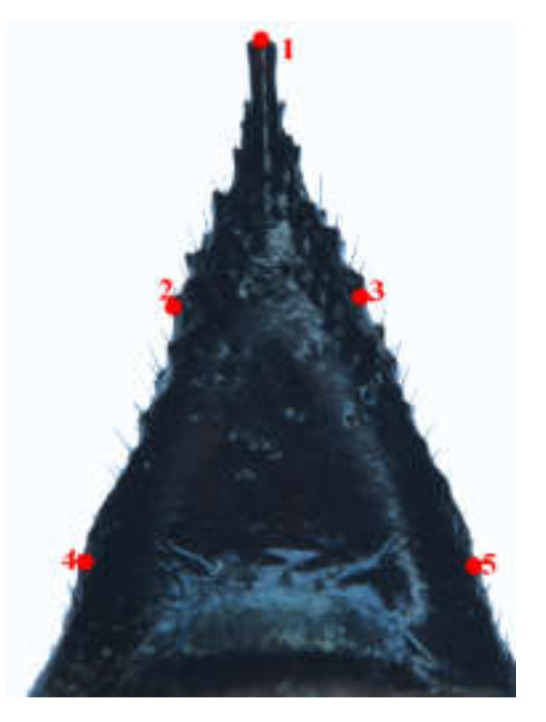
Position of 5 type Џ landmarks on the cornus of *Sirex* considered in the geometric morphometric analysis.

**Figure 4 insects-11-00084-f004:**
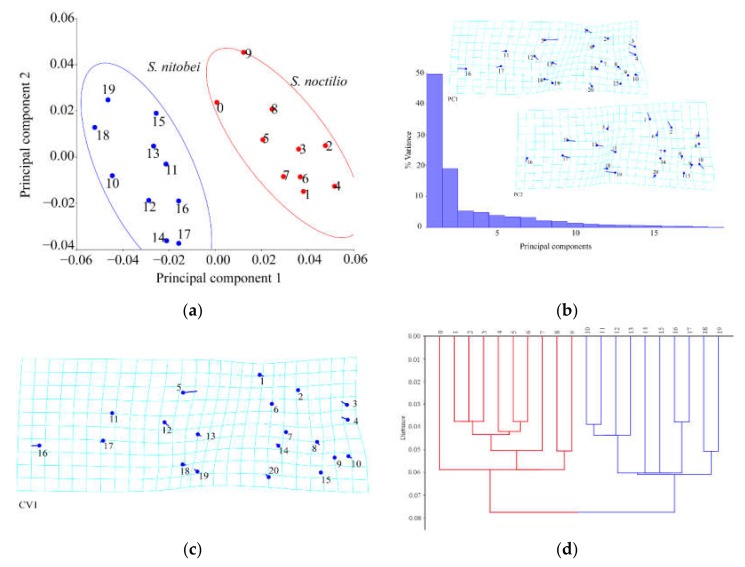
Shape variables of the female forewings of *S. noctilio* and *S. nitobei* (**a**) principal component analysis-(**b**) Transformation grids for visualizing a shape change (for the first two principal component, in this case)-(**c**) The Tps grids of Canonical Variate-(**d**) Phenogram of cluster analysis.

**Figure 5 insects-11-00084-f005:**
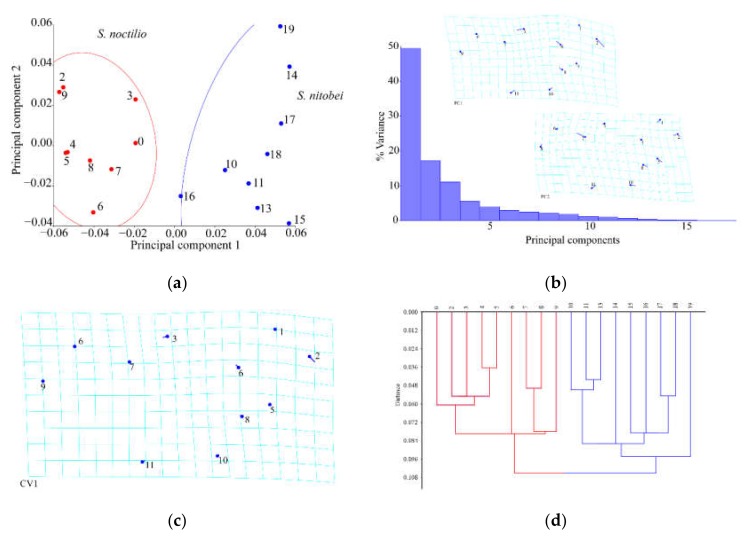
Shape variables of the female hindwings of *S. noctilio* and *S. nitobei* (**a**) principal component analysis-(**b**) Transformation grids for visualizing a shape change (for the first two principal component, in this case)-(**c**) The Tps grids of Canonical Variate-(**d**) Phenogram of cluster analysis.

**Figure 6 insects-11-00084-f006:**
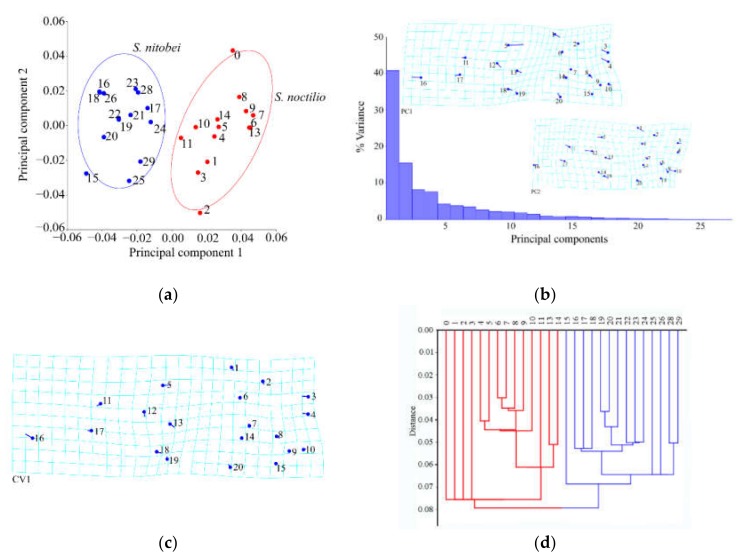
Shape variables of the male hindwings of *S. noctilio* and *S. nitobei* (**a**) principal component analysis-(**b**) Transformation grids for visualizing a shape change (for the first two principal component, in this case)-(**c**) The Tps grids of Canonical Variate-(**d**) Phenogram of cluster analysis.

**Figure 7 insects-11-00084-f007:**
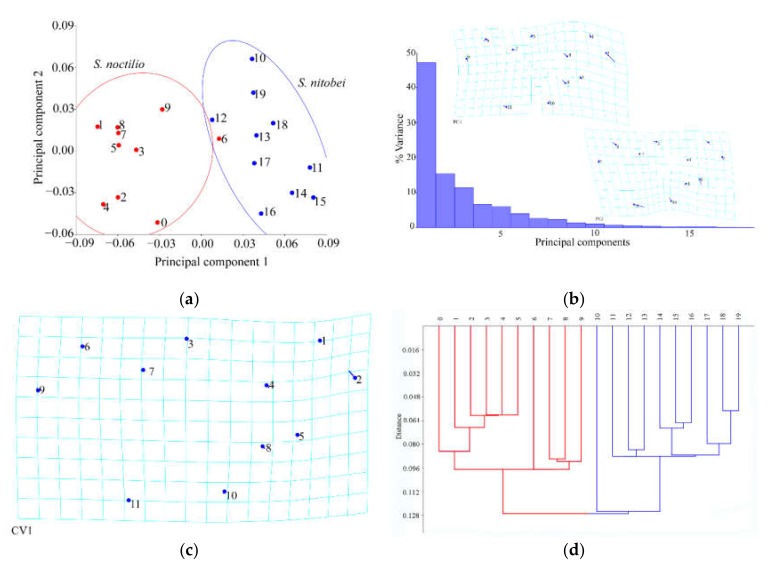
Shape variables of the male hindwings of *S. noctilio* and *S. nitobei* (**a**) principal component analysis-(**b**) Transformation grids for visualizing a shape change (for the first two principal component, in this case)-(**c**) The Tps grids of Canonical Variate-(**d**) Phenogram of cluster analysis.

**Figure 8 insects-11-00084-f008:**
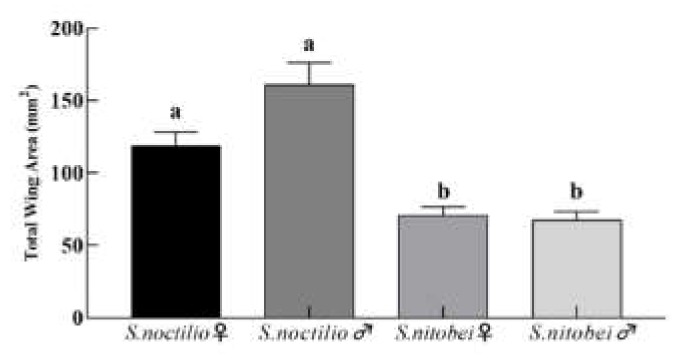
Comparison of total forewing area among two *Sirex*. Different letters indicate significant differences in total wing area among woodwasps within each sex, based on Tukey–Kramer’s multiple comparison tests at the 5% significance level.

**Figure 9 insects-11-00084-f009:**
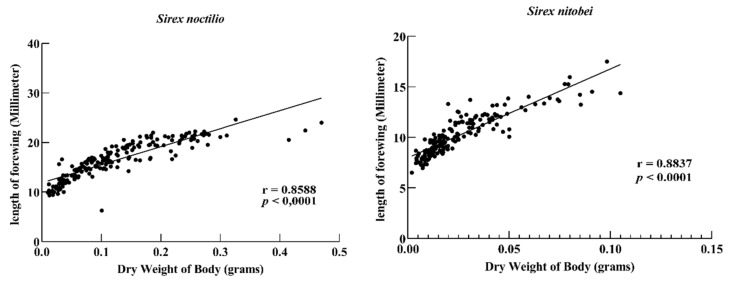
Relationship between the dry weight of body and the length of forewing (left: *Sirex noctilio*; right: *Sirex nitobei*).

**Figure 10 insects-11-00084-f010:**
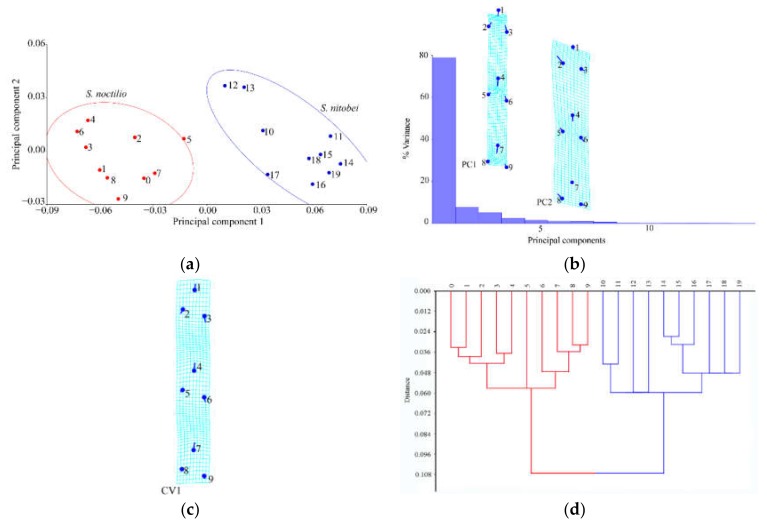
Shape variables of the ovipositor of *S. noctilio* and *S. nitobei* (**a**) principal component analysis-(**b**) Transformation grids for visualizing a shape change (for the first two principal components, in this case)-(**c**) The Tps grids of Canonical Variate-(**d**) Phenogram of cluster analysis.

**Figure 11 insects-11-00084-f011:**
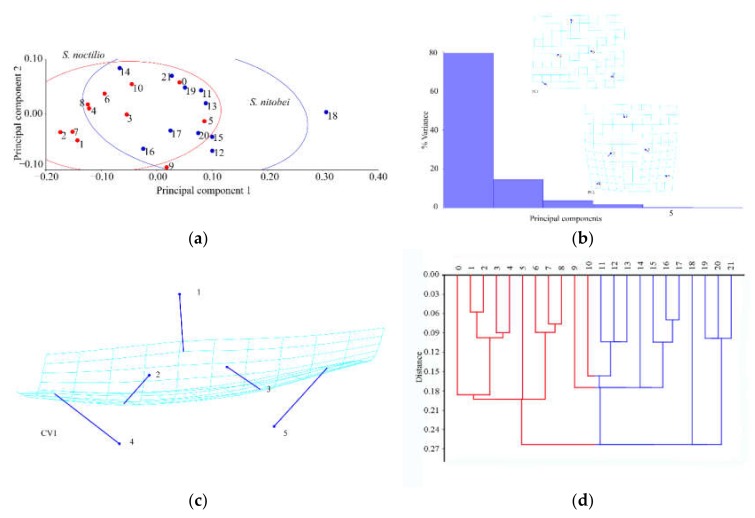
Shape variables of the cornus of *S. noctilio* and *S. nitobei* (**a**) principal component analysis-(**b**) Transformation grids for visualizing a shape change (for the first two principal components, in this case)-(**c**) The Tps grids of Canonical Variate-(**d**) Phenogram of cluster analysis.

**Table 1 insects-11-00084-t001:** The *Sirex* specimens collection information in this study.

Species	Family	Host	Number		Collecting Date
			Female	Male	
*Sirex noctilio*	Siricidae	*Pinus sylvestris* var. *mongolica* Litv	20	20	2016/07/15
*Sirex nitobei*	Siricidae	*Pinus sylvestris* var. *mongolica* Litv	20	20	2016/09/05

**Table 2 insects-11-00084-t002:** Differences in the female wings shapes among the species of Mahalanobis distances (left) & Procrustes distances (right): *p*-values (above); distances between populations (below) (10,000 permutation rounds).

		Forewings				Hindwings			
		*S. noctilio*	*S. nitobei*	*S. noctilio*	*S. nitobei*	*S. noctilio*	*S. nitobei*	*S. noctilio*	*S. nitobei*
		Mahalanobis distances		Procrustes distances		Mahalanobis distances		Procrustes distances	
*S. noctilio*	*p*-values	-	<0.0001	-	<0.0001	-	0.0001	-	<0.0001
*S. nitobei*	distances	11.0555	-	0.0611	-	23.0325	-	0.0830	-

**Table 3 insects-11-00084-t003:** Landmarks of forewing (according to veins nomenclature system by Ross (1937).

No.	Junctions of Veins	No.	Junctions of Veins
1	2A (anal veins) and 2cu-a (cubitoanal crossvein)	11	M and 3r-m (radiomedial crossvein)
2	1A and 2A	12	M and 2m-cu
3	2A and a (anal crossvein)	13	M and 2r-m
4	1A and a	14	M and 1m-cu
5	Cu (cubitus) and 2m-cu (mediocubital crossveins)	15	Rs (radial sector) and M
6	Cu and 2cu-a	16	R1 (radius) and Rs2
7	Cu and 1m-cu	17	Rs2 and 3r-m
8	Cu and 1cu-a	18	Rs and 2r-m
9	M (media) and Cu	19	Rs and 2r (radial crossvein)
10	M and Cu1	20	Rs and 1r

**Table 4 insects-11-00084-t004:** Landmarks of hindwing (according to veins nomenclature system by Ross (1937).

No.	Junctions of Veins	No.	Junctions of Veins
1	1A and cu-a	7	M and m-cu
2	1A and 2A	8	M and 1r-m
3	Cu and m-cu	9	Rs and 2r-m
4	Cu and cu-a	10	Sc (subcosta) and Rs
5	M and Cu	11	C (costa) and R1
6	M and 2r-m		

**Table 5 insects-11-00084-t005:** Differences in the male wings shapes among the species of Mahalanobis distances (left) & Procrustes distances (right): *p*-values (above); distances between populations (below) (10,000 permutation rounds).

		Forewings					Hindwings				
		*S. noctilio*	*S. nitobei*	*S. noctilio*	*S. nitobei*	*S. noctilio*		*S. nitobei*		*S. noctilio*	*S. nitobei*
		Mahalanobis distances		Procrustes distances			Mahalanobis distances			Procrustes distances	
*S. noctilio*	*p*-values	-	<0.0001	-	<0.0001		-		<0.0001	-	<0.0001
*S. nitobei*	distances	10.7298	-	0.0581	-		23.4236		-	0.0828	-

**Table 6 insects-11-00084-t006:** Differences in the ovipositor shapes among the species of Mahalanobis distances (left) & Procrustes distances (right): *p*-values (above); distances between populations (below) (10,000 permutation rounds).

		Ovipositor			
		*S. noctilio*	*S. nitobei*	*S. noctilio*	*S. nitobei*
		Mahalanobis distances		Procrustes distances	
*S. noctilio*	*p*-values	-	<0.0001	-	<0.0001
*S. nitobei*	distances	12.0359	-	0.0986	-

**Table 7 insects-11-00084-t007:** Differences in the cornus shapes among the species of Mahalanobis distances (left) & Procrustes distances (right): p-values (above); distances between populations (below) (10,000 permutation rounds).

		Cornus			
		*S. noctilio*	*S. nitobei*	*S. noctilio*	*S. nitobei*
		Mahalanobis distances		Procrustes distances	
*S. noctilio*	*p*-values	-	0.0067	-	0.0006
*S. nitobei*	distances	1.6984	-	0.1390	-

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
