# Peer review of "Comparison of Wing, Ovipositor, and Cornus Morphologies between Sirex noctilio and Sirex nitobei Using Geometric Morphometrics"

_insects, 2020, doi:10.3390/insects11020084_

Round 1

Reviewer 1 Report

The manuscript titled ‘Comparison of wing, ovipositor, and cornus morphologies between Sirex noctilio and Sirex nitobei using geometric morphometrics’ provides information on identification of these invasive and native species. As it is difficult to identify the wing vein characteristics (which are also unstable) of these two woodwasps with naked eye, the authors used geometric morphometrics.

The results or manuscript concludes that there is species variation in wing veins of these two woodwasps. Specifically, differences in forewings and pits on the ovipositor between the invasive S. noctilio and the native S. nitobei. Information from this study can be used to identify two Sirex species.

The manuscript is well written, with many details. The authors clearly explained the identification analysis and figures. However, the font size of the numbers in some figures need to be increased. For example (line 229, 231), numbers in PCA graphs, numbers in transformation and Tps grids, and in phenograms. Fix this in all other related figures.

Other minor changes are specified below:

Line 242 - Delete ‘.’ Lines 405, 408 – Italicize the scientific names

Reviewer 2 Report

Thank you for the chance to review this interesting paper. I think it adds to Sirex knowledge, and presents some interesting options for comparing closely related species in the future.

I do not find the paper very clear about the overall goal and significance is based on the introduction and discussion. I think the authors could do a better job outlining the goal of the research - is there an actual problem being addressed, or is this just demonstrating that these methods can indeed find differences between species? If only the latter then that's ok, but I was left wondering if I am missing some grander point. For example, the abstract suggests that this study "assists in clarifying the taxonomic uncertainties of Siricidae" - how, exactly? There did not seem to be any uncertainties highlighted for these two species.

The discussion meanders a bit, and can be hard to follow because of translation difficulties. For instance, the sentence on lines 328-330... are you saying that because they have the same host plant, any differences in morphology observed must be genetic rather than environmental? If so, I had to read this a couple of times to make sense of it. If not, then this definitely needs to be re-written. The sentence immediately after suggests the results allowed you to identify "possible evolutionary trends"... I didn't find any discussion of this in the paper, so maybe this is just indicating that this techniques could be used to identify trends. If so, that is another example of just a little editing needed to improve readability. 

I find that in general the discussion would benefit from a once-over by a native English speaker in collaboration with the authors just to ensure there is no confusion. There are few words in the paper you should replace - e.g. 'reproductive' instead of 'spawning', that will be captured with a final english edit. 

Other comments - figures in the final version need to be larger. It's impossible to see landmarks in the cornus image, for instance, and the red dots in general are difficult to see. 

You collected 40 individuals of each species, but only measured ten of each for the cornus and ovipositor? This is a small subsample. How did you select these individuals, and how did you ensure that the selection process did not impact your results? Did you also only measure ten for the wings? This seems to be the case based on the figures, but that is not clear from the text.

Table 1 makes it seem like all 40 S. noctilio were collected on 15 June, and all 40 S. nitobei were collected on 5 Sep. Is this the case, or are these dates supposed to be the range over which all 80 species were collected?

For wings with damaged outlines that you corrected (line 175), please describe how this was done without inadvertently altering the outline shape.

I added a scanned version of the pdf with marked edits. I had been doing it more to help me during the review, but hopefully some of these will be readable and useful. 

Reviewer 3 Report

In their manuscript titled "Comparison of wing, ovipositor, and cornus morphologis between Sirex noctilio and Sirex nitobei using geometric morphometrics", Wang et al. explore the use of morphometric analyses to identify two closely related species that are pests of pine trees in northeastern China.  For the most part the manuscript is well written, with clear and easy to follow methods and results, and I believe after some minor editing, the manuscript will make a nice addition to Insects.  The primary concern (though that might be too strong of a word) that I have is that the authors need to do a better job of convincing the reader that their approach (geometric morphometrics) is of use to researchers and for biosecurity.  Specifically, the authors state that they use leg and abdomen color differences to identify 20 samples of each species, and then they use those sorted specimens to perform their geometric morphometric analyses.  While they state that characters like color are problematic, it seems that some discussion of how reliable the character was in this system is necessary.  Where there any individuals where the color patterns said one thing, but the morphometrics said another?  Similarly, there seems to be rather large differences in the overall size of the forewings between the two species.  If leg color and wing size are accurate, then why do morphometric analyses need to be performed?  Some discussion could clarify this and add greatly to the manuscript.

Minor points:

The Abstract could use extensive editing.  The body of the text is well written, and easy to follow, but the Abstract seems to have been thrown together.  Please edit.

Introduction

Line 59. Start new paragraph with "The wood boring wasp..."

Line 64. Correct "damaged"

Lines 64-67. Edit for clarity

Lines 67-68. Please explain what differential evolutionary process you're referring to (or remove sentence).

Lines 72-72. Please explain why "unstable" wing patterns allows for morphometric analyses as this doesn’t seem intuitive.

Lines 74-75.  Please move the first sentence of this paragraph so that it comes after the current second sentence.

Methods

Figure 1.  Why is the hindwing not made binary as well?

Line 176-177. Remove this sentence (beginning with "With this approach...")

Results

Figures.  All. Please make the labels (both the axis labels and the sample labels) larger. As is, they are illegible.

Figures.  Phenograms.  Could the two species be colored (or labeled some other way) differently?  It would be helpful to know which clusters correspond to which species.

Figure 7. At least one individual seems to be misidentified by the male hindwing.  Is this true?  It would be helpful to compare this result to the other measurements and comment whether abdomen/leg color is the source of this misidentification or whether male hindwing is simply not as good for differentiating these two species.

Figure 11.  Similarly, the cornus seems like a poor indicator of species.  If the phenograms were labeled that would be more clear. 

Discussion

Lines 328-330.  I'm not clear what this sentence is trying to inform the reader.

Lines 336-338.  I agree that this type of interspecific variation could be problematic, but could the authors please SHOW whether it was for them?  It looks like (based on the results) that color patterns did a very good job differentiating species.
